# *Paracoccidioides* HSP90 Can Be Found in the Cell Surface and is a Target for Antibodies with Therapeutic Potential

**DOI:** 10.3390/jof6040193

**Published:** 2020-09-28

**Authors:** Ágata Nogueira D’Áurea Moura, Diane Sthefany Lima de Oliveira, Verenice Paredes, Letícia Barboza Rocha, Fabiana Freire Mende de Oliveira, Gustavo Meirelles Lessa, Juan Fernando Riasco-Palacios, Arturo Casadevall, Patrícia Albuquerque, Maria Sueli Soares Felipe, Roxane Maria Fontes Piazza, André Moraes Nicola

**Affiliations:** 1Institute of Tropical Medicine, Department of Dermatology, Faculty of Medicine, University of São Paulo, São Paulo 05403-000, SP, Brazil; agatandmoura@gmail.com; 2Institute of Biological Sciences, University of Brasília, Brasília 70910-900, DF, Brazil; diane.biomed@gmail.com (D.S.L.d.O.); msueliunb@gmail.com (M.S.S.F.); 3Karan Technologies Research and Development, Brasília 70632-200, DF, Brazil; vereniceph@gmail.com; 4Faculty of Medicine, University of Brasília, Brasília 70910-900, DF, Brazil; fafabi.oliveira@gmail.com (F.F.M.d.O.); gm.lessa@hotmail.com (G.M.L.); juanfdoriascos@gmail.com (J.F.R.-P.); 5Bacteriology Laboratory, Butantan Institute, São Paulo 05503-900, SP, Brazil; leticia.rocha@butantan.gov.br (L.B.R.); Roxane.piazza@butantan.gov.br (R.M.F.P.); 6Department of Molecular Microbiology & Immunology, Johns Hopkins Bloomberg School of Public Health, Baltimore, MD 21205, USA; acasade1@jhu.edu; 7Faculty of Ceilândia, University of Brasília, Brasília 72220-275, DF, Brazil; palbuquerque@unb.br; 8Graduate Program in Genomic Sciences and Biotechnology, Catholic University of Brasília, Brasília 70790-160, DF, Brazil

**Keywords:** monoclonal antibody, *Paracoccidioides* spp., HSP90

## Abstract

Paracoccidioidomycosis (PCM) is one of the most frequent systemic mycoses in Latin America. It affects mainly male rural workers in impoverished regions, and the therapy can last up to two years or use drugs that are very toxic. Given the need for novel safe and effective approaches to treat PCM, we have been developing monoclonal antibodies (mAbs) that could be used not only to block specific fungal targets, but also modulate the host’s antifungal immunity. In this work we show the generation of and promising results with an mAb against Heat Shock Protein (HSP)90, a molecular chaperone that is an important virulence factor in fungi. Using recombinant *Paracoccidioides lutzii* (Pb01) and *P. brasiliensis* (Pb18) HSP90 proteins produced in *E. coli*, we immunized mice and generated polyclonal antibodies and an IgG1 hybridoma mAb. The proteins were very immunogenic and both the polyclonal serum and mAb were used in immunofluorescence experiments, which showed binding of antibodies to the yeast cell surface. The mAb successfully opsonized *P. lutzii* and *P. brasiliensis* cells in co-incubations with J774.16 macrophage-like cells. Our results suggest that this mAb could serve as the basis for new immunotherapy regimens for PCM.

## 1. Introduction

Paracoccidioidomycosis (PCM) is an invasive fungal disease endemic to Latin America. It affects mostly poor rural workers and has a high medical and social impact due to its chronicity and frequent fibrotic sequelae, which lead to inability to work and poor quality of life [1]. Although PCM can be treated and cured, the available therapeutic tools are far from optimal. The drug of choice for severe cases, Amphotericin B (AmB), is very nephrotoxic [2] and requires hospitalization for intravenous use. Itraconazole and sulfonamides are also effective, but must be used for 9–24 months [1]. Furthermore, in vitro studies have shown that *Paracoccidioides* spp. can develop resistance to sulfonamides [3,4] and azoles [5].

Given the dire need for safer and more effective therapies, our group has been working on the development of new antifungal drugs. A prime target for these is the *Paracoccidioides* response to temperature increase, which is essential for the morphological conversion from mycelium to yeast and thus for the establishment of infection [6]. In thermodimorphic fungi, an increase in temperature such as the one between the environment and the human host induces the synthesis of Heat Shock Proteins (HSPs) [7], a family of evolutionarily conserved molecules [8]. Among these, HSP90 is essential for the folding, maturation and activation of approximately 10% of the yeast proteome [9]. We have previously shown that HSP90 inhibitors radicicol and geldanamycin are lethal to *P. brasiliensis* [10]. Moreover, an antibody fragment targeting the *Candida albicans* HSP90, called efungumab, reached phase III of multicentric clinical trials [11] before being discontinued.

Since PCM is a debilitating and potentially fatal disease when untreated [12], and in attempting to remedy the problems with the usual treatments, the study of recombinant antibody-based drugs has intensified. In this work we generated and evaluated polyclonal and monoclonal antibodies against the *P. brasiliensis* and *P. lutzii* HSP90 proteins. These antibodies bound to the cell wall and were opsonic for *Paracoccidioides* spp., showing their potential usefulness as starting points for novel antifungal drugs.

## 2. Materials and Methods

### 2.1. Cell lines, Fungal Strains and Growth Conditions

*Escherichia coli* strain DH5α was used for intermediate cloning steps and BL21 DE3, for protein expression. The human monocyte cell line THP-1 was cultured in Roswell Park Memorial Institute (RPMI) 1640 medium, and the Human embryonic kidney (HEK293) cells in Freestyle F17 expression medium. Mouse macrophage-like cells J774.16 were cultured in Dulbecco’s modified Eagle’s medium (DMEM) (Media sourced from Gibco Life Technologies, Grand Island, NY). Cell lines were maintained at 37 °C, 5% CO_2_, and passaged every three days. The Pb01 (*P. lutzii*) and Pb18 (*P. brasiliensis*) isolates were maintained in Yeast Peptone Dextrose Agar medium (YPD), at 37 °C. Cultures no older than five days from the last passage were used to prepare inoculum in YPD Broth. Incubation was carried out for 5 days, at 37 °C/180 RPM. Strains of *Candida albicans* (SC3514) and *Cryptococcus neoformans* (H99) were maintained in Sabouraud Dextrose Agar (SDA) and cultivated in Sabouraud Dextrose Broth (SDB), under 180 RPM stirring, at 30 °C, being used after 24 and 48 h, respectively (Media sourced from BD Difco, Franklin Lakes, NJ).

### 2.2. Animals

BALB/C female mice, six to eight weeks old, were immunized with HSP90 recombinant proteins from Pb01 and Pb18. Four experimental groups with four animals each were inoculated to produce polyclonal antibodies, and two groups with five animals each were inoculated for monoclonal antibody generation. The use of animals in experimentation was approved by the Ethics Committee on Animal Use from the Catholic University of Brasilia (CEUA/UCB) under protocol number 018/13 on 21 May, 2013.

### 2.3. Plasmid Vectors

Hsp90 coding sequences from Pb01 and Pb18 were retrieved from the Broad Institute online database and chemically synthesized by Epoch Biosciences Inc (Washington, DC, USA). They were cloned separately by the company into the pET21a prokaryotic expression vector with the addition of poly-histidine tags.

### 2.4. Production and Purification of Recombinant Hsp90

*E. coli* strain BL21 DE3 was transformed with vectors containing the HSP90 genes. Protein induction was carried out in TB medium containing 0.5 mM isopropyl β-D-1-thiogalactopyranoside (IPTG) at 37 °C/250 RPM/5–6 h after the culture reached OD600 = 0.6–0.8. Cells were centrifuged at 8000× *g* for 20–30 min at 4 °C, and the pellet was resuspended in lysis buffer (10 mM imidazole, 20 mM phosphate, 500 mM NaCl, pH 8). Lysis was performed in Ultrasonic Cell Disruptor, using as parameters: pulse on of 10 s, pulse of 59 s, and amplitude of 60%. The supernatant was filtered through a 0.45 µm nylon membrane and loaded into an ÄKTA pure system for affinity chromatography purification with a HisTrap Chelating Sepharose High Performance column charged with nickel sulfate. Fractional elution was performed against a linear imidazole concentration gradient in 20–30 column volumes of a 20 mM phosphate/500 mM NaCl buffer. A second purification step was performed by gel filtration in the ÄKTA system using a Superdex 75 10/300 size exclusion column. All steps followed the supplier’s instructions (GE Life Sciences, Marlborough, MA, USA).

### 2.5. Protein Extracts

Fungal cell mass from 100 mL of Pb01, Pb18, SC 3514 and H99 cultures, cultivated as described above, were washed once in 1× PBS and three times in 10 mM Tris-HCl buffer, pH 7.4. Centrifugation steps between washes were at 8000 g, 4 °C for five minutes. Cells were resuspended in the same buffer, previously chilled and supplemented with protease inhibitor (SIGMAFAST Protease Inhibitor Cocktail Tablets, EDTA-Free). Glass beads (500 µm) were added and cells were lysed by three 1 min cycles of vortexing with 30 s intervals on ice. Cell debris were spun down at 3000 × *g* /4 °C/5 min and separated from the cytosolic soluble protein fraction in the supernatant. To obtain the surface protein fraction, pellets were washed three times with ice-cold water, and three times with solutions of decreasing salt concentration (NaCl 5%, 2% and 1%), supplemented with 1 mM phenylmethylsulfonylfluoride (PMSF). Pellets were resuspended in extraction buffer (50 mM Tris-HCl, pH 8.0; 0.1 M EDTA, 2% SDS and 10 mM dithiothreitol) and incubated at 100 °C for 10 min. The samples were then centrifuged at 17,000× *g* for 10 min, at 4 °C, to collect the supernatant. To extract soluble proteins from THP1 and J774.16 mammalian cells, the steps were performed with RIPA lysis and extraction buffer (Thermo Fisher Scientific, Rockford, IL, USA) plus protease inhibitors. Some experiments were made with HEK cells rather than THP1, in this case mammalian cells were pelleted at 200× *g*, washed once with PBS and resuspended in cold RIPA buffer plus 1 mM phenylmethylsulfonyl fluoride, then vortexed for 30 s, incubated on ice for 30 min and centrifuged at 14,000× *g*, 10 min. Total protein concentration in all extracts was estimated by the Bradford or BCA methods. Samples were kept at −20 °C prior to use.

### 2.6. TCA Precipitation

When necessary, fungal cytosolic and surface extracts were concentrated by trichloroacetic acid (TCA) precipitation prior to electrophoresis. Cold TCA (1 g/L) was added at a 1:10 ratio to a volume of each fungal extract sample corresponding to 100 µg of total protein, followed by 30 min of incubation on ice. Samples were then centrifuged at 15,000 g for five min at 4 °C, and protein pellets were washed with 1 mL of a cold solution of 80% ethanol 20% 50 mM Tris-HCl, pH 8.0, centrifuged for 1 min as before, air-dried for 10 min after removing the supernatant, and resuspended by vortexing in 5 µL of 4× SDS-PAGE loading buffer plus 15 µL of 150 mM Tris-HCl, pH 8.5.

### 2.7. SDS-PAGE and Western Blot

Protein samples in loading buffer were boiled for five minutes, loaded and resolved into 12% SDS-polyacrylamide gels under reducing conditions. Gels were either stained with Coomassie Blue solution to document the protein profile of samples, or blotted onto nitrocellulose or PVDF membranes using a Trans-Blot SD Semi-Dry Electrophoretic Transfer Cell (Bio-Rad, Hercules, CA, USA) at 15 V for 15 min. After blotting, membranes were blocked overnight at 4 °C with PBS 5% skim milk, then incubated with appropriated antibodies for detection. An alkaline phosphatase (AP) conjugated anti-poly-histidine mAb (Sigma-Aldrich, St. Louis, MO, USA) was used to detect recombinant HSP90 protein, and produced anti-HSP90 polyclonal serum or anti-HSP90 mAb (clone 4D11 culture supernatant at 70 µg/mL, quantified by ELISA) were used for detection of fungal HSP90 proteins. As secondary antibodies, a mix of AP-conjugated mAbs against murine IgG, IgA and IgM was used to detect the binding of anti-HSP90 polyclonal sera, and a horseradish peroxidase (HRP)-conjugated mAb against mouse-IgG1 to detect the binding of anti-HSP90 mAb 4D11. Between each step, the membrane was washed three times for 5 min with PBS Tween 20 0.05%. For documentation, membranes incubated with AP-conjugated antibodies were rinsed in AP buffer, and the chromogenic signal was revealed with 5-bromo-4-chloro-3-indolyl-phosphate (BCIP) and nitroblue tetrazolium (NBT) reagent; the chemiluminescent signal of membranes incubated with HRP-conjugated antibody was generated using the SuperSignal West Pico Plus substrate (Thermo Fisher Scientific, Rockford, IL， USA) and captured with the ChemiDoc XRS+ System (BioRad, Hercules, CA, USA).

### 2.8. Production of Polyclonal and Monoclonal Antibodies

To produce polyclonal antibodies, mice received 50 µL with 50 μg of heterologous Hsp90 emulsified in complete Freund’s adjuvant, followed by three applications with the same amount of HSP90 in incomplete Freund’s adjuvant. Injections were performed subcutaneously at the lower abdomen and spaced at two-week intervals. Immunization groups were as follows: group 1 were immunized against the protein from Pb01, group 2 against the protein from Pb18, and group 3 against HSP90 antigens of both species inoculated alternately (both in complete adjuvant–1st and 2nd immunizations–and incomplete adjuvant). A fourth group was used as a negative control, receiving only PBS in Freund’s adjuvant. ELISA analyses showed higher antibody titers for group 3, and the same immunization method was then used to obtain lymphocytes. Lymphocytes collected from popliteal lymph nodes were fused with murine myeloma cells SP2/O-Ag14 (ATCC) to generate hybridomas as previously described [13].

### 2.9. Selection and Clone Isolation

Polyclonal sera, obtained from the retro-orbital sinus of the immunized mice, or hybridoma culture supernatants were tested against recombinant Pb01 and Pb18 HSP90, or fungal protein extracts, to confirm chaperone recognition and characterize the mAbs by isotyping. Plates with 96 wells were incubated with the antigens and blocked with 5% skim milk. The primary antibodies were then added to plates. Detection was made with an IgG/IgA/IgM anti-mouse complex, or specific anti-immunoglobulin antibodies for isotyping, conjugated to alkaline phosphatase. The substrate p-Nitrophenyl Phosphate (pNPP) was measured at 405 nm in a spectrophotometer. Incubations were performed at 37 °C/1 h, and washings with 0.05% Tween in PBS between each step. Pure cultures of mAb-producing hybridomas were obtained by limiting dilution.

### 2.10. 4D11 mAb Serum-Free Production and Purification

Clonal hybridoma cells growing in DMEM-10% SFB were centrifuged at 250× *g* for five minutes, resuspended in a 1:9 mixture of the previous medium and CD Hybridoma AGT (Gibco Life Technologies, Grand Island, NY, USA) serum-free medium, followed by incubation at 37 °C and 5% CO_2_. After two days, the cells were centrifuged and resuspended in pure CD Hybridoma AGT medium, seeded at a density of 5 × 10^5^ cells/mL in 1 L conical flasks with 200 mL of medium, and incubated at 37 °C, 5% CO_2_, 80 RPM. The suspensions were harvested after 6 days, when the cells were removed by centrifugation and the supernatant vacuum filtered through a 0.22 µm membrane. Purification of the 4D11 mAb was carried out by affinity chromatography using a Protein A/Protein G GraviTrap 1-mL column (GE Life Sciences, Marlborough, MA, USA), following the manufacturer’s instructions. Bound mAb was eluted with a 0.1 M glycine-HCl solution, pH 2.7, and the eluate was immediately neutralized with 1 M Tris-HCl pH 9.0 buffer. The final concentration of purified of the 4D11 mAb was determined by ELISA, following the protocol described above and using the equation given by the standard curve of the purified myeloma IgG1/k (MOPC21).

### 2.11. Immunofluorescence

After 2 days of 200 mL culture, the cells of the Pb01, Pb18, SC3514 and H99 isolates were centrifuged at 6000× *g*, 5 min, and washed with 1 mL PBS, for counting in a hemocytometer. 1 × 10^7^ cells were fixed with 1 mL of 4% formaldehyde for one hour at 25 °C. They were then centrifuged and incubated in 1 mL of methanol at −20 °C for permeabilization during 30 min at 4 °C. After being washed, incubation was carried out with 1 mL of the polyclonal sera or monoclonal antibodies against HSP90, the sera diluted 1:100 and the mAb 4D11 diluted to 50 µg/mL. Cells were washed and incubated for 1 h at 37 °C or overnight at 4 °C, with 1 mL of 1 µg/mL of anti-mouse IgG secondary antibody conjugated with AlexaFluor 488 (Invitrogen, Carlsbad, CA, USA). After washing, the pellets were resuspended in 20 µL of ProLong™ Gold Antifade mounting medium (Thermo Fisher Scientific, Rockford, IL, USA). Samples were then mounted onto glass slides and documented in a Zeiss Axio Observer Z1 microscope equipped with a 63×, 1.4 NA objective, a cooled CCD camera and a motorized stage. Z-series were collected and processed by an iterative limited deconvolution algorithm in the Zeiss ZEN software. Three-dimensional reconstructions were made with ImageJ and VOXX software. Images were processed with Adobe Photoshop and Adobe Illustrator. No non-linear modifications were made to them.

### 2.12. Phagocytosis Assay

J774.16 macrophage-like cells were plated onto 96-well plates at a density of 5 × 10^4^ cells/well. After 24 h, 10^5^ Pb01 or Pb18 yeasts were added per well in the presence of the 4D11 mAb (10 µg/mL) for two hours at 37 °C and 5% CO_2_. The cells were then washed with PBS, fixed with methanol at −20 °C for 30 min and stained with panoptic kit. Pictures were captured in the same microscope as before. Macrophages were counted in three fields per well, containing at least 100 cells total. The phagocytosis rate was calculated as the number of macrophages with internalized fungal cells divided by the total number of macrophages counted.

### 2.13. Statistical Analysis

Graph Pad Prism 7.1 software was used for statistical analysis of the phagocytosis assay. Datasets were compared by one-way ANOVA and a Tukey’s post-test.

## 3. Results

### 3.1. Paracoccidioides Spp. HSP90 Proteins are Highly Immunogenic

Induction for at least 3 h, in LB medium, with 0.5 mM IPTG, at 37 °C, was the optimal condition to produce the HSP90 protein from both *Paracoccidioides* species (Figure 1). We were unable to achieve total purity of either HSP90 proteins. The extra bands observed in stained SDS-PAGE gels were not recognized by the anti-His antibody, indicating they are either HSP90 degradation products that do not contain the his-tag or unrelated proteins (Figure 2). Given that the relative amount of the contaminants was very low, we used these suspensions to immunize mice. Sera from mice that were immunized with *P. brasiliensis* or *P. lutzii* recombinant HSP90 were able to recognize and bind HSP90, as were sera from animals that were immunized with a mixture of both proteins. Sera obtained by alternated immunization against proteins from both fungi yielded the highest titer (1:256,000).

### 3.2. Production of Monoclonal Antibodies Against Paracoccidioides HSP90

After fusion, selection, expansion on feeder layers, and primary screening, 56 clones that yielded a significant ELISA signal were transferred to 24-well plates for further expansion. Nine hybridomas with good cell growth (1G12, 2B3, 2B4, 2B5, 2C2, 2C4, 2D2, 4D10 and 4D11) were expanded, and antibodies from their supernatants were isotype defined by ELISA. Only three clones, from 2B3, 2B5 and 4D11 wells, presented antibodies of a single subtype (IgG1). Cells derived from other wells produced a mix of isotypes, so the limiting dilution, expansion, screening and isotyping steps were repeated. A second round of isotyping resulted in two clones derived from the 2C2 hybridoma (2C2/1G12 and 2C2/2G5), both producing an IgG2b mAb, and two from the 4D11 hybridoma (4D11/1E5 and 4D11/2D4), both producing an IgG1 mAb. Analysis by ELISA suggested that 4D11/1E5 and 2C2/1G12 (lost throughout the development of the project due contamination) had a higher affinity for the target proteins. The remaining hybridoma (2C2/1G12) supernatant, as well as 4D11, were shown to bind HSP90 on the surface of *Paracoccidioides* spp. yeast cells by epifluorescence microscopy (Appendix A).

### 3.3. HSP90 is Located in the Cell Wall of Paracoccidioides spp.

Sera with the highest polyclonal antibody titer were able to recognize the HSP90 band from cytosolic protein extracts of all the fungi tested (*P. brasiliensis*, *P.luzii*, *C. albicans* and *C. neoformans*). When cell surface protein extracts were used instead, detection only happened in the *Paracoccidioides* samples. The heat shock proteins from J774.16 and THP1 protein extracts were not recognized (Appendix A). Purified 4D11 mAb at 70 µg/mL was able to recognize HSP90 in all samples, both in cytosolic and surface extracts (100 µg/well). 4D11 mAb did not bind proteins from HEK cells (Figure 3).

Epifluorescence microscopy with deconvolution showed a diffuse signal in whole fungal cells (Pb01 and Pb18) when treated with polyclonal antisera (Appendix A) or 4D11 monoclonal antibody (Figure 4) against HSP90. Some autofluorescence could be seen in the negative controls, but the fluorescence intensity in cells treated with the antibody was higher. Tests made with SC3514 and H99 yeast cells showed a weaker signal in comparison to *Paracoccidioides* strains. The negative control for the experiments with polyclonal antibodies included sera from animals that were injected only with PBS plus adjuvant, showing that no non-specific antibody binding occurred.

### 3.4. Antibodies to HSP90 Effectively Opsonize Paracoccidioides spp.

*Paracoccidiodes* spp. yeast cells that had been opsonized with the 4D11 mAb were phagocytosed more efficiently than non-opsonized ones. The mean phagocytosis rate for opsonized Pb18 was 59%, and for opsonized Pb01 it was 71%, while the rates for non-opsonized cells were 11% and 44%, respectively. In contrast, SC3514 and H99 cells were not phagocytosed more efficiently when opsonized with the 4D11 mAb (Figure 5).

## 4. Discussion

The treatment of PCM, an important infectious cause of chronic lung disease and the main non-opportunistic, systemic mycosis in Latin America [12,14], must usually be protracted to control the clinical manifestations and to avoid relapses. The patient must remain in treatment and follow-up until the cure criteria are achieved, based on clinical, radiological and serological parameters [15]. In this work we have succeeded in the production of polyclonal antisera and an IgG1 isotype mAb (4D11) against *Paracoccidioides* spp. HSP90, with potential as a new tool for PCM therapy. The mAb was produced from murine cells by hybridoma technology and specifically recognized and bound to the HSP90 chaperone in vitro and in *Paracoccidioides* spp. cells. It successfully opsonized *Paracoccidioides* spp. yeast cells, indicating its potential for PCM immunotherapy. The mAb also bound to *C. albicans* and *C. neoformans* yeast cells, albeit with a weaker signal that suggests lower affinity. It was not effective in opsonizing these species, though, perhaps due to a combination of lower affinity and the cryptococcal capsule hindering FcγR access to the mAbs.

We detected HSP90 both in the cytosol and in the *Paracoccidioides* spp. cell wall by immunofluorescence microscopy, using both polyclonal sera and mAb; images from cells labeled with sera were clearer than those stained with the mAb, but both were nevertheless positive when compared with the negative controls. Moreover, Western blot experiments with soluble and cell wall protein extract confirmed the result. HSP90 is a protein commonly found in the cytosol and abundant in all eukaryotic cells [16]. In fungi such as *Aspergillus fumigatus* and *Saccharomyces cerevisiae* under normal growth conditions, HSP90 has been found widely distributed in the cytosol and in some cytoplasmic structures. Under stress situations, it has been observed to relocate to the nucleus or the cell wall [16,17,18]. Cell wall HSP90 distribution was also found in *C. albicans* by Western blotting against cell surface protein extracts and by Flow cytometry, both using polyclonal anti-HSP90. The chaperone was found as 82-, 72- and 47-kDa fragments in the cytosol [19]. In another study, a proteomic analysis of the *C. albicans* plasma membrane identified HSP90 as an associated membrane protein [20]. The present work is the first to report the location of HSP90 protein on the cell surface of *Paracoccidioides* spp. Since it is an essential protein for the performance of many cellular functions, and, therefore, a potential target, its location on the outer surface of fungal pathogens means it can interact with therapeutic agents.

Efungumab, originally called Mycograb, is a recombinant human, single-chain variable fragment (scFv) against *C. albicans* HSP90. It was studied as therapy for systemic candidiasis in combination with current antifungals, and produced by NeuTec Pharma, a subsidiary of Novartis AG. It was developed from coding cDNA of anti-HSP90 antibodies from patients who had invasive candidiasis and recovered. This immunoglobulin derivative does not penetrate fungal cells and interacts solely with extracellular or membrane-bound HSP90 [11], highlighting the importance of the target molecule location for antibody-based strategies. Unlike mAbs developed by hybridoma technology, the fragment is devoid of the Fc domain, which was claimed to enable its effectiveness even in immunocompromised patients [2]. However, due to the heterogeneity of the molecule—batches showed structural inconsistencies caused by self-aggregation—the commercial production for this purpose was then discontinued.

Other studies with murine mAbs have shown good responses in experiments against fungal antigens in vitro and in vivo. Guimarães et al. succeeded in producing mAbs against *Histoplasma capsulatum* HSP60 by the hybridoma technique [21]. These antibodies, like those produced in this work targeting HSP90, did not recognize mammalian chaperones, which suggests that cross-reactivity against protein of the infected host is not a concern. These anti-HSP60 mAbs are also protective against *P. lutzii*, reducing the fungal load in infected mouse lungs. These findings indicate that mAbs produced against an evolutionarily preserved protein from a fungal species may exhibit reactivity and have therapeutic effect against mycoses caused by a range of species. This is in line with our finding that the HSP90 mAb we produced against the chaperone of *Paracoccidiodes* spp. was able to recognize proteins of *C. albicans* and *C. neoformans*. Very promising results were also found in tests with an IgG2b to gp43 [22] and an IgG1 to gp70 [23], both against proteins from *P. brasiliensis*. The antibodies showed a protective effect in passive immunization of infected mice, improving lung lesions and reducing the fungal load [22,23]. More recently, fully human mAbs generated by cloning VH and VL sequences from single B cells were shown to bind to *Candida* spp. cell surfaces, opsonizing them and protecting mice infected with *C. albicans* [24].

Monoclonal antibodies are isotypically homogeneous and specific to one epitope. The identification mAbs protective against fungi contribute to the development of immunotherapeutic tools and to the characterization of the molecules they recognize, helping explain their importance in virulence and pathogenicity of the corresponding organism. However, murine mAbs as the ones we report in the present paper fall short of being directly useful as therapeutic agents due to the fact that they are recognized by the human immune system as foreign molecules, which can lead to the production of human anti-mouse antibodies (HAMA) that neutralize them or lead to the formation of immune complexes [25]. Thus, murine mAbs serve mainly the purpose of enabling in vivo tests of their behavior in a mouse model, generating results that suggest whether they might be effective enough to warrant attempts at antibody humanization with a view to clinical trials. The 4D11 mAb we report will be investigated with that purpose in mind.

## Figures and Tables

**Figure 1 jof-06-00193-f001:**
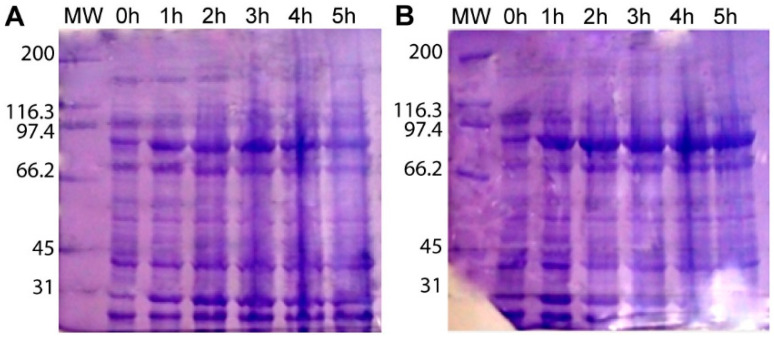
Recombinant Heat Shock Protein (HSP)90 production in *E. coli* BL21 DE3. Whole-cell protein extract from *E. coli* cultures producing recombinant HSP90 from *P. lutzii* (**A**) or *P. brasiliensis* (**B**) at different times after induction. MW: Molecular Weight standard.

**Figure 2 jof-06-00193-f002:**
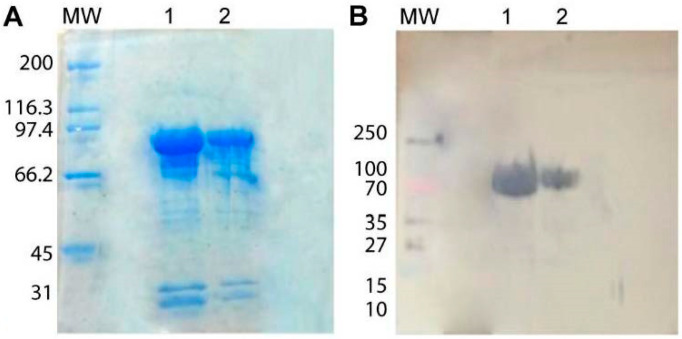
Recombinant HSP90 protein purification, SDS-PAGE and Western blotting. (**A**) SDS-PAGE of affinity-purified recombinant *P. lutzii* and *P. brasiliensis* HSP90. (**B**) Western Blot with chromogenic detection of anti-polyhistidine mAb binding. Lane (1) *P. lutzii* recombinant HSP90 (2) *P. brasiliensis* recombinant HSP90. MW: Molecular Weight standard.

**Figure 3 jof-06-00193-f003:**
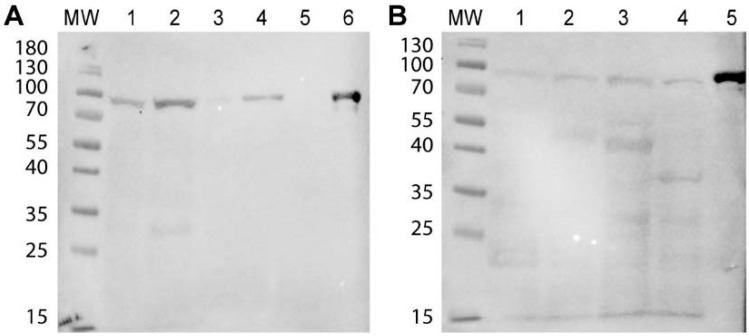
HSP90 detection in fungal and mammalian protein extracts with mAb 4D11. (**A**) Western Blot with 4D11 monoclonal antibody (mAb) and cell surface protein extracts from *P. lutzii* (1), *P. brasiliensis* (2), *C. albicans* (3) and *C. neoformans* (4), Human embryonic kidney (HEK)293 human whole-cell protein extract (5) and recombinant *P. lutzii* HSP90 (6). (**B**) Similar experiment, but with cytosolic protein extracts from *P. lutzii* (1), *P. brasiliensis* (2), *C. albicans* (3) and *C. neoformans* (4) and recombinant *P. lutzii* HSP90 (5). MW: Molecular Weight standard.

**Figure 4 jof-06-00193-f004:**
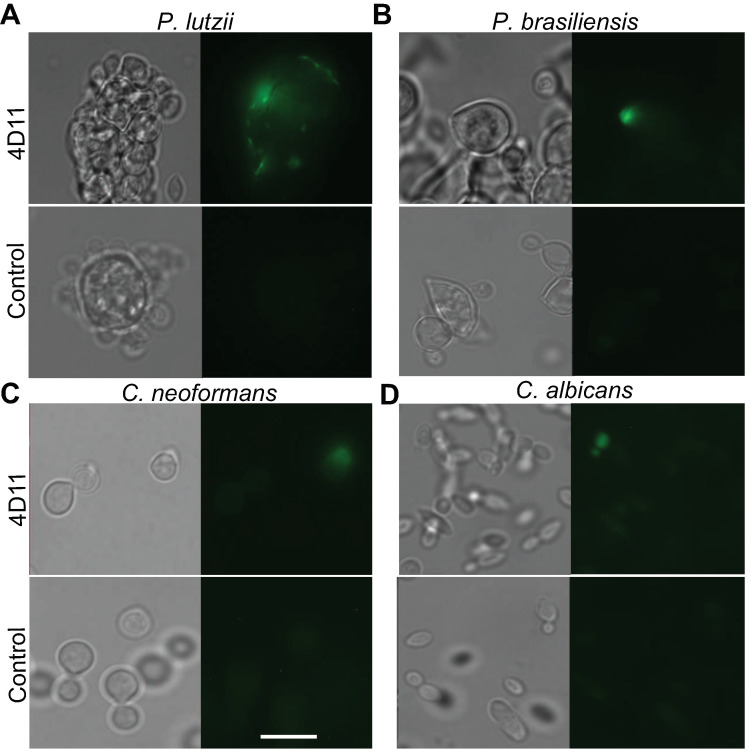
HSP90 immunolocalization in yeast cells with monoclonal antibodies. Yeast cells were incubated with the 4D11 mAb or PBS (Negative Control). Detection was made with an Alexa Fluor 488-conjugated secondary Anti-IgG mAb (Invitrogen). Protein was detected in the surface and in the cytosol. Cells were fixed with 4% formaldehyde. (**A**) *P. lutzii* (**B**) *P. brasiliensis* (**C**) *C. neoformans* (**D**) *C. albicans* Left quadrants: (DIC) Differential interference contrast. Right quadrants: (IF) Immunofluorescence. Scale bar: 10 µm.

**Figure 5 jof-06-00193-f005:**
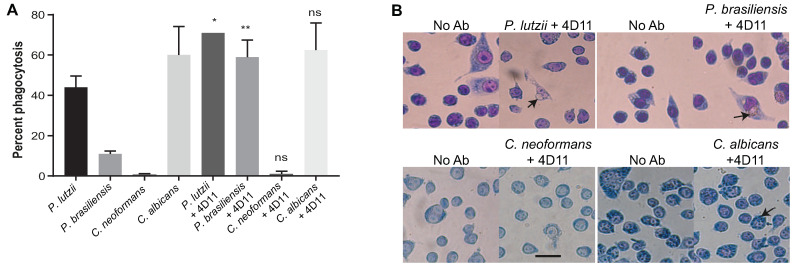
Phagocytosis assays with mAb 4D11. (**A**) Macrophage-like J774.16 cells were incubated with 4D11-opsonized and non-opsonized fungi. After 2 h, non-internalized fungi were washed away and the cells were stained and imaged. The y-axis shows the percentage of macrophage-like cells with ingested yeast cells. Bars represent the average percentage of macrophages with at least one internalized fungal cell. Datasets resulted from two independent experiments and were analyzed by one-way ANOVA and Tukey’s post-test (* *p* < 0.05; ** *p* < 0.001). (**B**) Representative images for phagocytosis assays with *P. lutzii* and *P. brasiliensis*. Left quadrants: Negative control–non-opsonized yeast cells. Right quadrants: Yeast cells opsonized with 4D11. Arrows indicate macrophages with internalized fungi. Scale bar: 10 µm.

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
