# Peer review of "Paracoccidioides HSP90 Can Be Found in the Cell Surface and Is a Target for Antibodies with Therapeutic Potential"

_jof, 2020, doi:10.3390/jof6040193_

Round 1
Reviewer 1 Report
In this manuscript, authors present a set of antibodies (polyclonal and monoclonal) binding HSP90 of P. brasiliensis and P. lutzii. It is very important to have this type of research published for fungal infections.
I would recommend the authors to be careful as some of the pathogens names need to be in italics.
Also, it would be good the authors discuss about why their antibodies seem to identify HSP90 in Cryptococcus in WB images but no differences in phagocytosis assays.
In addition, localising fluorescence images are very clear for polyclonal antisera and hybridoma supernatants (supplementary materials), but not very clear for Pb18 stained with monoclonal antibody+Alexa 488 (Fig 4). Do the authors have better images? In figure 4 it seems the localisation of this antibody is good only for Pb01 (for Pb18, H99 and SC3514 signals seem equally low). Also, authors need to add analysis of the MFI to the fluorescence images mainly as they referred to it in text but no quantitative data was shown.
Authors need to add a IgG1 control for their monoclonal experiments (PBS only control are not enough) and rule out that the effects seen are HSP90 antibodies specific.
In discussion, I recommend the authors to review and include Rudkin et al. Nat commun 2018 Dec 11;9(1):5288. doi: 10.1038/s41467-018-07738-1.
Author Response
In this manuscript, authors present a set of antibodies (polyclonal and monoclonal) binding HSP90 of P. brasiliensis and P. lutzii. It is very important to have this type of research published for fungal infections.
Response: We thank the reviewer for the positive comments.
I would recommend the authors to be careful as some of the pathogens names need to be in italics.
Response: We appreciate the important observation. We have thoroughly re-checked the manuscript to ensure that all scientific names are in italics.
Also, it would be good the authors discuss about why their antibodies seem to identify HSP90 in Cryptococcus in WB images but no differences in phagocytosis assays.
Response: We agree with the recommendation. We believe the explanation is a mix of lower affinity to the antigens that were not used in immunizing mice and the cryptococcal capsule blocking FcγR access to the antibodies that bind to HSP90 on the cell wall. We have added this interpretation to the Discussion, lines 307-311.
In addition, localising fluorescence images are very clear for polyclonal antisera and hybridoma supernatants (supplementary materials), but not very clear for Pb18 stained with monoclonal antibody+Alexa 488 (Fig 4). Do the authors have better images? In figure 4 it seems the localisation of this antibody is good only for Pb01 (for Pb18, H99 and SC3514 signals seem equally low).
Response: These are the best images we were able to collect in this experiment. We agree they are not as clear or pretty as some of the images in the supplemental figures, but we were able to see in each one of them cells that were positive, with a pattern that was not observable in the negative controls.
Also, authors need to add analysis of the MFI to the fluorescence images mainly as they referred to it in text but no quantitative data was shown.
Response: We agree with the reviewer. We used terminology that is usually applied to quantitative image analysis, such as “mean fluorescence intensity” and “signal-to-noise ratio”. However, we did not perform quantitative image analysis, and do not believe it is necessary or even helpful in this case. We have thus revised the text to make this clearer (lines 266 and 267).
Authors need to add a IgG1 control for their monoclonal experiments (PBS only control are not enough) and rule out that the effects seen are HSP90 antibodies specific.
Response: We have indeed not included an irrelevant, isotype-matched monoclonal antibody in the negative control for the experiments with monoclonal antibodies. The experiment shown in figure S2, however, includes a negative control in which the cells were pre-incubated with sera from mice that were injected with adjuvant plus PBS. No signal was observable above background in these cells, proving that they do not bind antibodies non-specifically.
In discussion, I recommend the authors to review and include Rudkin et al. Nat commun 2018 Dec 11;9(1):5288. doi: 10.1038/s41467-018-07738-1.
Response: We accept the suggestion. The study by Rudkin et al. was reviewed and discussed in the Discussion session, lines 348-350.
Reviewer 2 Report
Paracoccidioides HSP90 can be found in the cell surface and is a target for antibodies with therapeutic potential
This is a well presented manuscript. Please see below some comments.
- Please review the use of italics across the document, for example Paracoccidioides in the title, and other species on results section.
- Please check last paragraph of induction. That text more than a study objective description, looks like a conclusion.
- The main goal of this study was to find MoAb with potential use on PCM treatment. Have this MoAb any potential double use? immunodiagnostics assays? In case of an affirmative response, please can you add a comment about this in the conclusion section as study future directions.
Author Response
This is a well presented manuscript. Please see below some comments.
Response: We thank the reviewer for the positive comment.
Please review the use of italics across the document, for example Paracoccidioidesin the title, and other species on results section.
Response: We appreciate the important observation. We have thoroughly re-checked the manuscript to ensure that all scientific names are in italics.
Please check last paragraph of introduction. That text more than a study objective description, looks like a conclusion.
Response: We agree and have reformulated this paragraph (lines 67-70).
The main goal of this study was to find MoAb with potential use on PCM treatment. Have this MoAb any potential double use? immunodiagnostics assays? In case of an affirmative response, please can you add a comment about this in the conclusion section as study future directions.
Response: We thank for the great and relevant comment. This antibody could indeed have some use in diagnostics for fungal diseases, given that HSP90 is a highly expressed and immunodominant antigen. However, as we do not show in this manuscript any experimental evidence of its usefulness in diagnostics, we preferred not to speculate without a solid base.
Round 2
Reviewer 1 Report
Thank you to the authors for replying and welcoming the feedback. I don't have any other comment more than a suggestion that the authors might want to declare in their discussion that they don't have better images for their monoclonal antibodies but that the pattern they saw was different from the negative controls and they didn't performed quantitative image analysis as they don't believe it's necessary or helpful. Same with the rationale behind not including the isotope-matched control.
Author Response
Thank you to the authors for replying and welcoming the feedback. I don't have any other comment more than a suggestion that the authors might want to declare in their discussion that they don't have better images for their monoclonal antibodies but that the pattern they saw was different from the negative controls and they didn't performed quantitative image analysis as they don't believe it's necessary or helpful. Same with the rationale behind not including the isotope-matched control.
Response: We thank the reviewer for the suggestions. We have modified two paragraphs in the Results (lines 262-264) and Discussion (lines 307 to 311) accordingly.